# Evaluation of Different LiDAR Technologies for the Documentation of Forgotten Cultural Heritage under Forest Environments

**DOI:** 10.3390/s22166314

**Published:** 2022-08-22

**Authors:** Miguel Ángel Maté-González, Vincenzo Di Pietra, Marco Piras

**Affiliations:** 1Department of Environment, Land and Infrastructure Engineering, Politecnico di Torino, 10129 Torino, Italy; 2Department of Topographic and Cartography Engineering, Escuela Técnica Superior de Ingenieros en Topografía, Geodesia y Cartografía, Universidad Politécnica de Madrid, Mercator 2, 28031 Madrid, Spain; 3Department of Cartographic and Land Engineering, Higher Polytechnic School of Ávila, Universidad de Salamanca, Hornos Caleros 50, 05003 Ávila, Spain

**Keywords:** LiDAR, terrestrial laser scanner, mobile mapping systems, airborne LiDAR sensor, unmanned aerial systems, cultural heritage, accuracy analysis, point cloud analysis

## Abstract

In the present work, three LiDAR technologies (Faro Focus 3D X130—Terrestrial Laser Scanner, TLS-, Kaarta Stencil 2–16—Mobile mapping system, MMS-, and DJI Zenmuse L1—Airborne LiDAR sensor, ALS-) have been tested and compared in order to assess the performances in surveying built heritage in vegetated areas. Each of the mentioned devices has their limits of usability, and different methods to capture and generate 3D point clouds need to be applied. In addition, it has been necessary to apply a methodology to be able to position all the point clouds in the same reference system. While the TLS scans and the MMS data have been geo-referenced using a set of vertical markers and sphere measured by a GNSS receiver in RTK mode, the ALS model has been geo-referenced by the GNSS receiver integrated in the unmanned aerial system (UAS), which presents different characteristics and accuracies. The resulting point clouds have been analyzed and compared, focusing attention on the number of points acquired by the different systems, the density, and the nearest neighbor distance.

## 1. Introduction

Architectural heritage is a resource of multiple dimensions (cultural, social, territorial, and economic) that greatly enriches and protects the societies in which the sites are located [1]. These heritages, symbols of regions, cities, and towns, are linked to their social sentiment and cultural identity and are often the basis for the development of activities on which their economy is based, such as tourism [2]. Monumental heritage is a reflection of the history of a territory, capable of tracing the passage of different civilizations. Its value is unquestionable, but its conservation and maintenance are not always feasible. They are very fragile elements that have suffered the effects and consequences of historical and natural events that have modified, altered, and, in the worst of cases, even destroyed them [3]. The conservation of this built heritage is one of the aspects that any advanced society must inevitably address. Today, apart from the deterioration due to the passage of time and the impact of meteorological agents and the effects of climate change, these assets are exposed to other constant threats, such as: (a) their abandonment (due to depopulation or lack of funds for their maintenance) or loss of functionality; (b) destructive actions towards these assets (vandalism and armed conflicts); (c) accidental destructive events (such as the fire that happened at Notre Dame Cathedral in Paris); (d) other types of hazards that are difficult to predict, such as natural catastrophes, earthquakes, among others; (e) threats linked to human activities, such as poor planning, management, or maintenance of these assets (uncontrolled tourism, corrective repairs, etc.) [4]. All these factors make the protection, management, research, dissemination, improvement, conservation, and safeguarding of architectural heritage at a global level a highly complex task [4].

Considering these circumstances, different international organizations and institutions (UNESCO or ICOMOS, among other international bodies), together with national and regional administrations, have promoted international charters [5,6], norms, and laws that attempt to deal with this problem. In almost all of them, apart from speaking of the values and importance of cultural heritage for today’s society and the need to preserve this historical legacy of past generations, they promote research and knowledge transfer from other fields of science, engineering, and other disciplines for the adaptation or creation of new methodologies and techniques of analysis, which favor and improve the current methods of intervention, conservation, and management of cultural heritage. 

Among this range of techniques and methodologies that improve studies and research on cultural heritage, geomatics plays a key role in capturing, processing, analyzing, interpreting, modeling, and disseminating digital geospatial information [3,7,8,9,10,11]. These sciences and technologies are particularly useful in the cultural heritage sector as they provide information on the current state of heritage assets and make it possible to highlight and quantify the changes that these assets may undergo in space and time [7,8,9,10,11,12,13,14].

Currently, the documentation, conservation, restoration, and dissemination actions carried out in the cultural heritage field avail the use of 3D recording strategies that allow the creation of digital replicas of assets in an accurate and fast way. These 3D recording strategies provide important information with metric properties of cultural heritage geometrically, structurally, dimensionally, and figuratively [3,7,8,9,10,11,12,13,14]. Thus, we have managed to improve graphic representation resources with more detailed planimetry and with the option of generating a 3D model from which we can obtain longitudinal and transversal profiles, virtual tours, calculation of surfaces, volumes, orthophotographs, monitoring of the object under study to check its deterioration, etc. [3,7,8,9,10,11,12,13,14]. Products can be used to support in-depth analysis of the architectural, engineering, and artistic features of the objects. In this context, 3D recording strategies have become indispensable tools and methodologies when carrying out studies on assets, conservation or restoration work, or their dissemination [12,13,14].

Nowadays, it is possible to find numerous investigations and works in which these techniques are applied in architectures of cultural and historical relevance. These heritage assets, in most cases, are located in cities or towns and their surroundings [7,8,9,10,11,12,13,14], where around 55% of the world’s population is concentrated [15]. However, it is rare to find projects dealing with the management and safeguarding of cultural heritage located in remote rural areas, such as mountain areas or formerly inhabited places, which are currently abandoned. As a general rule, the population in the 1960s left these territories (in which there were hardly any services) and moved to the cities, looking for new opportunities for their families, taking advantage of the restructuring of large industries [16]. This led to the disappearance of many historic mountain villages, which based their economy on artisanal activities, such as the exploitation of wood or the extraction of raw materials from the subsoil (mining) [17]. At present, many of these villages and the remains of the industrial craft activities they carried out are in a deplorable state of preservation (some of them have disappeared) and are invaded by vegetation and have been absorbed by wooded areas, making their 3D documentation very complicated. 

In recent years, the geomatics sector has undergone a significant transformation, which has made it possible to improve 3D digitizing methods, opening up the possibility of researching and progressing in the documentation of these complex scenarios. Firstly, the improvement of static scanning systems and the appearance of new dynamic scanners, known as mobile mapping systems (MMSs) [10,12,13,14], have made it possible to speed up the reconstruction of complex scenarios with a simple walk. This is thanks to the combination of an inertial measurement unit (IMU), a LiDAR scanning system, and the application of SLAM (simultaneous location and mapping) techniques and visual odometry for the processing of these data. MMSs are mostly based on ROS (robotic operative system) being widely used in robotic navigation and in autonomous driving. These systems allow to perform dead-reckoning position estimation without the necessity of a GNSS receiver and implement novel algorithms for registering point clouds and extracting maps [10,12,13,14]. In terms of data quality, these devices usually offer centimeter accuracy in contrast with the last generation terrestrial laser scanner (TLS), which reaches a millimeter or sub-centimeter level of accuracy [10,12,13,14]. Moreover, point cloud resolution depends on the acquisition rate, the distance to the object at any given time, and the number of passages. Although these devices are more suitable for indoor use, due to their higher productivity and efficiency than outdoor use, they have been successfully used for reconstructing outdoor scenarios, such as archaeological sites, churches, civil engineering elements, among others [10,12,13,14].

Secondly, the rise of commercial off-the-shelf (COTS) drones, together with a substantial improvement in their performance and the miniaturization of the electronic components, has allowed installing an airborne LiDAR sensor (ALS) on a flying platform, ensuring greater flight autonomy, precise positioning in flight (thanks to GNSS—global navigation satellite systems- or INS—inertial navigation system), and a lower cost compared to classic alternatives (such as satellite images or photogrammetric flights). 

In relation to the above, it may seem strange not to mention photogrammetric methods, which have revolutionized 3D documentation in cultural heritage [7,18,19,20]. It is true that new developments in photogrammetry (SFM—structure from motion-, computer vision… [21]) have made it possible that almost anyone with little knowledge in the field, with an image capturing device (cell phone, tablet, camera, or commercial drone with camera), following a simple protocol in data collection, and with the help of commercial or free software [21], is able to obtain point clouds or 3D models of objects or simple scenarios [3,22,23]. However, it is more than demonstrated that, to date, these techniques, which use images (such as input), cannot provide information beyond what appears in them [21]. Since we are dealing with cultural properties covered by vegetation and absorbed by wooded areas, if we use these methods, we would be documenting all the biological parts that cover and alter the heritage element and not the element itself. 

This research aims to compare the results obtained using several active systems of massive data capture: a TLS, a WMMS, and a drone ALS, which allow the documentation of the cultural heritage even if it is covered by biological invasions [24]. Consequently, it is necessary to analyze in depth the possibilities offered by these systems for the documentation of this type of scenario, as well as their limitations, and to evaluate their advantages and disadvantages as a possible step toward strengthening the management of endangered cultural heritage. As can be seen, Section 2 introduces the study case and Section 3 defines the materials and methods used to digitalize it. Section 4 describes the experimental results obtained andincludes the discussion, and, finally, some conclusions obtained from the present work are drawn within Section 5.

## 2. Case Study

### 2.1. Monte Pietraborga, Trana, Provincia di Torino, Piemonte, Italy

The Monte Pietraborga is a low-altitude small mountain system that is part of the Eastern Italian Alps (Central Cozie Alps). It is composed of several ridges and peaks (the highest being about 926 m). It is located at the head of the Sangone Valley, within the municipalities of Trana (mostly), Sangano, and Piossasco in the Metropolitan City of Torino in the Piedmont region. It lies a few kilometers southwest of Torino (Figure 1). From the plain where the city of Turin is located, the skyline of Monte Pietraborga and its neighbor Monte San Giorgio (837 m high at its highest peak) can be seen. Its advanced position with respect to the plain represents an important visual reference point for the surrounding area. 

The rocks that compose the mountain are part of the Massiccio Ultrabasico di Lanzo, a geological formation of ancient origin. The predominant rocks, the peridotite, are very rich in magnesium, and this also influences the nature of the soils formed by their degradation. Thanks to this contribution of substrates, the vegetation present in the mountain is quite lush, highlighting the chestnut, oak, and hazel, among other vegetation.

In Monte Pietraborga, several historical villages are almost uninhabited or completely uninhabited, among which stands out the village of Pratovigero, located on the northwest slope. The access to this village is from Trana, by a poorly preserved dirt road. Other villages are Prese de Piossasco and Prese di Sangano. These are made up of small groups of houses located at different points on the southern slope of Monte Pietraborga. Their access is very complex due to the poor preservation of the roads, some of them nowadays blocked by obstacles, such as fallen trees. Historical documents confirm that some of these centers have been inhabited since the 10th century.

Apart from the aforementioned population centers, there are other heritage elements of historical importance on Monte Pietraborga, such as: (i) the Cappella Madonna della Neve, built in 1700 (in a poor state of preservation); (ii) several fountains and springs; (iii) an area with remains of Celtic vestiges (dolmens and menhirs arranged in a circle, known as Sito dei Celti, and dating back to the period between 4000 and 2800 BC). 

Most of the people who lived in this mountainous environment were engaged in primary sector activities: (i) they had domestic livestock and cultivated the fields to feed the animals (rye, fodder, etc.) and for their consumption (wheat, potatoes, turnips); (ii) they were collectors of the fruits provided by the mountain flora, such as: hazelnuts, walnuts, chestnuts, and other types of cultivated fruits, such as pears, apples, and cherries; (iii) or were dedicated to the exploitation of the resources and raw materials of the area (wood, coal, serpentinite). 

Since the beginning of the twentieth century, due to the absence of fundamental services, the inhabitants of these territories abandoned them and moved to the city of Turin in search of new opportunities for their families, taking advantage of the restructuring of the large industries [17,25,26]. This exodus worsened after the end of the Second World War. 

Today, most of the heritage assets of this territory are abandoned and in a poor state of conservation, mostly invaded by vegetation or absorbed by wooded areas.

#### Study Area Selection

In order to select a suitable site for testing the different geomatic sensors, a study was carried out to locate all the buildings or constructive elements covered by vegetation located on Monte Pietraborga. The data used for this study are based on: (i) the vector layer of the buildings registered in the Cadastre of the Province of Turin (Figure 2a, elements in red); (ii) and the high-resolution raster layers of the forest cover density, provided by the COPERNICUS service of the European Union (https://land.copernicus.eu/, accessed on 22 July 2022) (Figure 2b). The first step that has been carried out is the creation of the centroids of the vector layer of the buildings registered in the Cadastre. Subsequently, the extracted points were assigned the value of the raster layer of the tree cover density. Finally, a selection of points was made to ensure that they were covered by more than 70% of the tree cover density (points in yellow in Figure 2). 

Among those points, the site that best fits the scope of our research has been selected after a survey in the field. Figure 3 shows the selected heritage element. It is a historical building with a typical construction of the territory of Monte Pietraborga architecture. It is located near the village of Pratovigero. At present, this building is abandoned and in ruins. As can be seen, the forest has covered it entirely, and part of it is covered by the surrounding vegetation.

## 3. Material and Methods

In order to evaluate the use of different laser scanner technologies for the specific purpose of documenting a submerged cultural heritage in an extremely vegetated environment, three different sensors among the most widely used in the market today were employed. The abandoned building analyzed in this study was the subject of both a terrestrial and aerial multi-sensor topographic measurement campaign. Specifically, the survey was performed at first with a terrestrial laser scanner (TLS), the Faro Focus 3D X130, a static point cloud acquisition tool based on the co-registration of different portions of the building acquired from different station points. Next, an MMS was used, the Kaarta Stencil, a handheld multi-sensor system consisting of a 16-beam laser scanner, a tracker-type camera, an inertial measurement unit (IMU), and a computation processor integrating advanced simultaneous localization and mapping (SLAM) algorithms. Finally, with the aim of validating its performance in an extreme environment, DJI’s new Zenmuse L1 ALS was used for an aerial platform laser survey. This LiDAR sensor combines data from an RGB sensor and the IMU unit in a stabilized 3-axis gimbal. While the TLS scans and the MMS data have been geo-referenced using a set of vertical markers and sphere measured by a GNSS receiver in RTK mode, the ALS model has been geo-referenced by the GNSS receiver integrated in the unmanned aerial system (UAS), which presents different characteristics and accuracies. Therefore, this incoherence in the final data must be considered in the following analysis. 

### 3.1. Methodological Approach for the Validation Analysis

The digital data of the architectural asset were evaluated according to typical metrics and different strategies used in the literature for point cloud analysis, using as ground truth the TLS cloud, which is, by sensor characteristics, the most accurate and dense (about an order of magnitude higher than other systems). To this scope, third-party software CloudCompare [27] was used to exploit different algorithms and tools for point cloud analysis and cloud-to-cloud comparison. 

Specifically, the number of points of each dataset has been reported for the original on-field survey and for a common portion of the environment (mainly the building and a small portion of the surroundings). The three LiDAR products have also been compared in terms of point density, defined as the number of points per unit in a given point cloud. The density is mainly determined by the acquisition techniques and post-processing filtering, the scanner resolution, and the scanning geometry [28]. In this research, the density was computed considering a volume defined by a 3D sphere with radius of 0.5 m, as expressed in (Equation (1)).
(1)$$\frac{N}{\frac{4}{3}\pi r^{3}}$$
where N equals the number of neighbors detected in a 3D sphere of 0.5 m radius around the given point.

Finally, a cloud-to-cloud distance analysis was performed on selected areas of the building, using as ground reference the TLS data and comparing both the MMS and the ALS results. The cloud-to-cloud distance was performed using the multiscale model to model cloud comparison (*M3C2*) algorithm [29], already implemented in CloudCompare. This approach estimates the discrepancies between the two point clouds defined in the same reference system. It is important to highlight that, when two clouds have been geo-referenced with coherent measurements and the same technique, the cloud-to-cloud comparison provides direct information about the performance of the acquisition system. On the other hand, when the geo-referencing procedure of the two clouds is different and different acquisition sensors provide the relative measurements, the cloud-to-cloud comparison could be affected by the geo-referencing accuracy. Therefore, considering the different geo-referencing procedures used in this work (GCP-based vs. direct RTK), the L1 data must be evaluated, referring not only to the measurement accuracy but also to the geo-referencing accuracy. In the first case, the *M3C2* was applied directly on the L1 data without any pre-processing (a), while, in the second case, the *M3C2* algorithm was preceded by a relative fine registering of L1 data to TLS data using an iterative closest point procedure (ICP) [30] (b). All the analyses to validate the three systems are summarized in Figure 4. The tests focused on the ability of these instruments to operate in extremely vegetated environments; in this regard, the methodological approaches used will be made explicit in the Results section.

#### Multiscale Model-to-Model Cloud Comparison (*M3C2*)

The *M3C2* algorithm allows a fast analysis for the calculation of the local mean distance of two large point clouds with irregular surfaces. For this purpose, it calculates a local mean cloud-to-cloud distance for a point in the reference cloud, called “core point”, by using a search cylinder projected along a locally oriented normal vector [31,32] (Figure 5).

The parameters used to calibrate the *M3C2* algorithm were selected following the literature [29,31,32,33]. Starting from the typical value selected for cultural heritage clouds, parameters have been tuned in an iterative approach. In the following table are reported the parameters from the first and last step of the calibration (Table 1).

### 3.2. Terrestrial Laser Scanner

To obtain a detailed model of the building and its immediate surroundings, a terrestrial laser scanner was used. More specifically, the lightweight terrestrial laser scanner Faro Focus 3D was used (Figure 6). This device measures distances using the principle of phase shift within a range of 0.6 to 120 m with a measurement rate of 976,000 points per second, an angular resolution of 0.009°, and a beam divergence of 0.19 mrad. Concerning its accuracy, this laser allows for capturing scenes with ±2 mm of precision under normal lighting conditions.

### 3.3. Mobile Mapping Systems

The MMS used for the present work was the Kaarta Stencil 2 mobile mapping system [34,35,36]. Kaarta Stencil 2 depends on LiDAR and IMU data for localization. Kaarta Stencil 2 is a stand-alone, lightweight SLAM instrument with an integrated system of mapping and real-time position estimation. In addition, it is a handheld of limited size, which allows quick and easy 3D mapping by hand made. The system uses Velodyne VLP-16 connected to a low-cost MEMS IMU and a processing computer for real-time mapping (Figure 7). VLP-16 has a 360° field of view with a 30° azimuthal opening with a band of 16 scan lines. A small tripod was used to support the system. The device can be connected to other sensors, such as a GNSS receiver or an external monitor with a wired USB connection (which allows to see the results in real-time). The data acquisitions were captured using the Kaarta Stencil 2 default configuration parameters, set in order to use the instrument in structured outdoor environments (Table 2). 

All these devices are carried by an operator, whose movement provides the third dimension required to generate the 3D point cloud. The 3D point cloud is generated by combining the information coming from the scanning head and the IMU sensor, using to this end the LOAM (LiDAR odometry and mapping) approach [34,35,36]. It is called LOAM: by finely using two algorithms running in parallel, it allows a practically real-time solution. An odometry algorithm estimates the velocity of the TLS, therefore correcting the errors in the point cloud while a mapping algorithm matches and registers the point cloud to create a map. It must be taken into consideration that the previous process is an incremental procedure in which each segment is aligned with respect to the previous one. The error accumulation derived from the incremental procedure is minimized by a global registration on the basis that the starting and ending points are the same (closed-loop solution).

This sensor accurately maps large indoor and outdoor spaces quickly and easily, with a range of up to 2–100 m with a LiDAR accuracy of ±30 mm (in outdoor environments). The data rate is 300,000 points per second up to 10 Hz. This device is also equipped with a camera that allows one to record a video while the laser is capturing the scene. The manufacturer ensures an accuracy of 1–3 cm for a 20-min scan, with the closing of a single loop.

### 3.4. Drone + LiDAR Sensor

The UAV chosen for this research is a DJI Matrice 300 RTK. This DJI device allows a wide range of sensors to be airborne, which will enable us to carry out different tasks in agriculture, construction, inspections, security, and mapping. The Matrice 300 RTK is among the best drones on the market thanks to its compatibility with multiple cameras, such as a high-resolution camera (45 megapixels) known as the DJI Zenmuse P1 camera or a LiDAR sensor (flight time) known as the DJI Zenmuse L1. The basic characteristics of this UAV are detailed in Table 3. 

The LiDAR sensor DJI Zenmuse L1 combines data from an RGB sensor and the IMU unit in a stabilized 3-axis gimbal, thus providing a true color point cloud from the RGB sensor (Figure 8). When used with the Matrice 300 RTK and DJI Terra software, the L1 forms a complete solution that provides real-time 3D data throughout the day, efficiently capturing the details of complex structures and providing highly accurate reconstructed models. Table 4 shows the technical characteristics of the sensor.

## 4. Results and Discussion

Previous to the data collection with the different LiDAR sensors, a network of registration spheres and target cards (horizontally and vertically) was placed along the scene with the aim of aligning the different point clouds generated in a common global coordinate system (Figure 9). Distributed throughout the scene, five markers have been placed horizontally on the ground in areas clear of wooded vegetation (Figure 9a), and the coordinates of these points have been obtained with a GNSS network real time kinematic (NRTK) survey for which a geodetic receiver, the multi-frequency, multi-constellation Leica GS18, has been used, receiving differential correction from the network service of continuous operating reference stations (CORS) provided by the Piedmont region (SPIN reference). The marker location accuracy was 1.2 cm in planimetry and 2.5 cm in altimetry. With these points and thanks to a total station survey (Leica MS50, Wetzlar, Germany), the global coordinates of the rest of the vertical target cards have been obtained thanks to the topographic radiation method (Figure 9b). These points have been obtained with millimeter precision.

All these control points have been used in two phases of the work: (i) as ground control points to check if the global reference system of the point cloud obtained by the airborne LiDAR sensor on the RTK drone coincides with the global reference system of the ground data acquisition; (ii) as control points to compare the data obtained from the TLS with respect to the data of the MMS; (iii) to align the different TLS scans with the classical point-based coregistration procedure. 

Regarding point (i), the estimated control points (marker, vertical target, and sphere) will allow us to check if the independent geo-referencing procedure of the RTK drone (cm-level of accuracy) is consistent with the GNSS survey on the ground. The RTK positioning performed directly on the fly by the DJI system is mandatory when no ground markers are visible from the drone point of view due to the massive presence of trees and vegetation. This is especially interesting in this work since the building is located in a wooded area where it is difficult to position ground control points.

Regarding the vertical targets, two of them have been fixed to the building, while the others have been placed on topographical tripods, where, later, the spheres were placed. It should be noted that the vertical targets central point coincides with the center of the sphere (with precision of ±1 mm) (Figure 9d). These devices have been calibrated in the topography laboratory of the Department of Environmental Engineering, Territory and Infrastructure (DIATI) of the Politecnico di Torino. In total, six registration spheres with a diameter of 200 mm have been placed along the scene, guaranteeing their visibility from different positions (Figure 9c).

### 4.1. TLS, MMS, and ALS Survey

In order to digitalize the entire building and its surroundings, nine scans have been made around the object, as shown in Figure 10 (yellow triangle). The high number of scans required for completing the building survey is due to the difficulties in observing directly the object in a woody area. The alignment of these scan stations was carried out by means of the target-based registration method. This method allows for aligning different point clouds through the use of geometrical features coming from artificial targets, such as planar targets or registration spheres. For the present study case, a target-based registration approach able to use the centroid of each registration sphere was used as a control point for the alignment between point clouds. Within this framework, the centroid of each sphere was extracted by the RANSAC shape detector algorithm [37]. As a result, it was possible to align all the point clouds with an accuracy of ±3 mm. The resulting point cloud is composed of 47,368,018 points. The field data acquisition of the nine scans took about 100 min (around 8 min per scan), and approximately 150 min for the post-processing of the merging of the different point clouds.

In addition, data acquisition was carried out with the Kaarta Stencil 2 mobile mapping system. Prior to the data acquisition with the MMS device, an on-site inspection was carried out with the aim of designing the most appropriate data acquisition protocol, taking into account the suggestions proposed by Di Filippo et al. [13], with the following statements standing out: (i) ensuring accessibility to all the areas, (ii) removing obstacles along the way, and (iii) planning a closed-loop in order to compensate for the error accumulation. During the data acquisition, a closed-loop path was followed with the aim of compensating possible error accumulations (Figure 10). In order to ensure a homogenous density of the point cloud, the walking speed was constant, paying special attention to transition areas.

Taking these considerations into account, a single loop was necessary to digitize the building and its surroundings, investing a total of 12 min.

Data acquisition with the Kaarta Stencil 2 device first required PC initialization. Along with this power-up, the IMU was started to establish the reference coordinate system, the tracking camera, and the LiDAR sensor. At the end of the acquisition phase with Kaarta Stencil 2, information about the configuration setting, the 3D point cloud characteristics, and the estimated trajectory is stored in a folder automatically created by the WMMS processer at every operation of the survey. Subsequently, precise processing of the field data was carried out in the laboratory, which took 40 min. The total resulting point cloud is composed of 63,256,457 points.

The UAS LiDAR-based survey was performed using a DJI Matrice 300 RTK drone piloted in manual mode to prevent collision in such a harsh environment. The real-time kinematic features allow to connect the drone to any RTK server via NTRIP protocol and, therefore, provide differential correction to the GNSS positioning system in real time. In a high vegetated environment, where GCP are not visible from the UAS, this feature allows direct geo-referencing with a high level of accuracy (few centimeters). This is the case in our study, where all the acquired data and derived products are geo-referenced directly without exploiting the GCPs under the canopy. 

Those data were acquired with the DJI Zenmuse L1 commercial system, a portable multi-sensor platform composed of a Livox LiDAR module, a CMOS RGB imaging sensor, and an IMU. As the L1 sensor supports multi-beam LiDAR acquisition, the parameters were set to acquire three return signals (maximum value) with a scanning frequency of 240 K/s to test its ability to penetrate vegetation. The raw data acquired are 5473 × 3648 RGB jpeg images, a proprietary data format for LiDAR data, inertial, and GNSS positioning data. The raw LiDAR data are converted in standard point cloud format (.las) with DJI Terra software, which also processes all the RTK and IMU data with the images to color the LiDAR point cloud. The obtained point cloud was composed of 8,768,243 points with associated RGB and intensity channels, as well as scan angle, time of acquisition, and return number. The UAS flight survey required about 15 min and the post-processing approximately 60 min. Figure 11 shows the obtained point cloud in RGB visualization and with signal return classification, from which is evident the difficulties to penetrate the vegetation in such a challenging scenario, where the building is beneath a dense forest and covered by the foliage. For this reason, the GCPs and the marker used in the previous analysis have not been used, and direct geo-referencing of the data has been performed. 

Analyzing the results of the raw data acquisition for each sensor, a higher number of cloud points are provided by the TLS Faro Focus, as expected. The Faro LiDAR sensor acquires 976,000 pts/sec with an angular resolution of 0.009 deg for each direction, the Velodyne VLP-16 integrated into the Kaarta Stencil acquires 300,000 pts/s with an angular resolution of 0.1 deg along the azimuth and 0.4 deg along the zenith, and, finally, the DJI L1 sensor acquires 240,000 pts/s. These characteristics are reflected in the number of points for each data type, as described in Figure 12. Regarding the density computed as the number of neighbors in a volume of 1 m^3^, the Kaarta Stencil is the most dense data acquired. The reason is due to the acquisition methodology. While the TLS point cloud is obtained co-registering a certain number of scans acquired from a fixed position, the MMS point cloud is obtained by the sum of the continuous acquisition process performed during the movement. This means that the density of the data could be increased by simply observing the same area more times during the walking path.

### 4.2. Data Comparison Analysis

#### 4.2.1. TLS vs. MMS

The data comparison analysis was performed in a selected portion of the LiDAR survey, in particular on the building covered by the vegetation. The first analysis compared the volume density of the TLS cloud with respect to the MMS one to investigate the performances of the two systems in the interest area. The density, expressed in pts/m^3^, has been computed from (Equation (1)), and the results are expressed in Table 5. Observing the density distribution evidences the difference between the two on-field data acquisition methods (Figure 13). As expected, the density values derived from the TLS survey present a high-tailed distribution, indicating more dense data in the overlapping areas and sparse data where the object is observed just from one scan position. On the other hand, the MMS has a Gaussian mixture distribution of the density value, with areas of high density, medium density, and lower density. Moreover, the mean density value is greater in the MMS cloud (Table 5). This is typical of an iterative LiDAR acquisition performed from a pedestrian, which cannot be constant in the velocity movement and, therefore, can cause different behaviors in the density and numerosity of the acquired points. 

In order to align the point cloud obtained by the MMS with the TLS point clouds, another target-based registration phase was carried out, using to this end the spheres (Figure 9). The root mean square error (RMSE) of this registration phase was 0.01 m. As stated before, the registration spheres were equipped on topographic tripods with special platforms that allow for geo-referencing the centroid of each sphere. Thanks to this, it was possible to geo-reference the point cloud obtained by means of a six-parameter Helmert transformation (three translations and three rotations), allowing for placing both LiDAR models in the same coordinate system. 

A point-to-point comparison was carried out to obtain a more in-depth evaluation of the potential of the MMS solution for mapping these spaces. During this stage, the multiscale model to model cloud comparison (*M3C2*) algorithm was used [29]. This approach allows for estimating the observed discrepancies between the MLS and TLS point clouds (Figure 14), and it is implemented in the open-source software CloudCompare v2.10 [27]. In order to obtain reliable results, the two point clouds were segmented by eliminating the non-common areas. The comparison of both point clouds provided an RMSE of 0.01 m (Figure 15), in line with the RMSE obtained during the alignment phase.

#### 4.2.2. TLS vs. ALS

As for the MMS data, the L1 point cloud was compared with the TLS point cloud using the latter as a reference. Firstly, the clouds were compared by number of points and density metrics. Point density has been computed in terms of the number of neighbors (NoN) detected in a 3D sphere of radius 0.5 m. The analysis is performed in the building area of the survey, identifying a common portion of the cloud (Figure 16). In this case, L1 data present a Gaussian density distribution with almost 30 times fewer points than the TLS data (Table 6).

Following the density analysis, a point-to-point comparison was carried out, exploiting again the *M3C2* algorithm implemented in CloudCompare software as homologous points cannot be defined. The *M3C2* distance computation calculates the changes between two point clouds along the normal direction of a mean surface at a scale consistent with the local surface variations. If the reference cloud is dense enough, then the nearest neighbor distance will be close to the “true” distance to the underlying surface. The first point-to-point comparison was performed considering the entire region of interest, i.e., the building under the canopy, where the TLS point cloud was used as a reference. Subsequently, point-to-point comparison was performed on different portions of the building (façade and planar section) to provide *M3C2* distance statistics also along the principal direction of the cloud. This allows to analyze the data acquired by the L1 sensor considering the vegetation density of the surrounding environment, which could differently affect the scanned surface. Moreover, dividing the analysis on a different portion allows to obtain a less error-prone statistical value. In Figure 17 is shown the different regions of interest. 

The geo-referenced TLS model and the L1 point cloud were both projected into the UTM cartographic representation (East, North UTMWGS84-32N ETRF2000) with elevation transformed from ellipsoidal height to orthometric height using regular interpolation grids provided by the Istituto Geografico Militare (IGM) (https://www.igmi.org/, accessed on 20 July 2022). Considering the different geo-referencing procedure (GCP-based vs. direct RTK), the L1 data must be evaluated referring not only to the measurements’ accuracy but also to the geo-referencing accuracy. In the first case, the *M3C2* was applied directly on the L1 data without any pre-processing (a), while, in the second case, the *M3C2* algorithm was preceded by a relative fine registering of L1 data to TLS data using an iterative closest point procedure (ICP) (b). For both analyses, a common portion of the surveyed area was selected and segmented. The parameters of the alignment process were optimized by minimizing the ICP alignment error. In particular, the number of iterations of the algorithm was fixed to the default value of 20, the optimum threshold for minimizing the root mean square (RMS) difference was found to be 10^−5^ cm, and the final overlap was set to 60%. With these parameters, 106,957 points out of 120,748 were selected for registration and the final RMS was 0.042 cm. The results of the analysis of the *M3C2* distance are statistically represented in Table 7 and visually reported in Figure 18.

The *M3C2* distance analysis (Table 8) clearly shows the potentiality of the L1 sensor to penetrate the vegetation and, at the same time, obtain an accurate survey. The point cloud obtained by the direct geo-referencing process presents some deviation with 50% of the points with a distance less than 24 cm. This value highly decreases after ICP computation reaching 1 cm of distance. In this case, 95% of the points have a distance less than 7 cm, which means an overall good performance of this system. It must be highlighted that the comparison has been made in a highly vegetated environment; therefore, these errors are affected by some noise related to the environment.

Finally, to provide insight regarding the penetration capability of the L1 sensor, the three signal returns have been analyzed again, computing the number of points and the cloud density for each beam (Table 9). As you can see, L1 is able to gather behind the canopy, also reaching a portion of the building, although with some void. 

## 5. Conclusions

In the present work, three LiDAR technologies (Faro Focus 3D X130 TLS, Kaarta Stencil 2–16 MMS, and DJI Zenmuse L1 ALS) have been tested and compared in order to assess the performances in surveying built heritage in vegetated areas. Each LiDAR surveying technique applies a different on-field and post-processing methodology due to the difference in the technological features. In the field, the rapidity and ease of data acquisition represent fundamental aspects, while, in post-processing, computing cost and automation of workflow are essential. In this regard, moving LiDAR technologies have a great advantage with respect to TLS: they can be easily deployed in the field and the data acquisition can be performed in a few minutes. Moreover, the processing cost is lower thanks to iterative co-registration algorithms, which allow scans to update in a dead reckoning fashion. In more detail, the MMSs are more versatile and can be mounted on aerial and ground vehicles or carried by a pedestrian, solving the problem of surveying in more scenarios. ALS, on the other hand, by taking advantage of the moving agent represented by the UAV, can cover large expanses of land, but only by returning an overhead view. In our specific scenario, i.e., a built heritage submerged by dense vegetation, the ALS demonstrated good penetration ability overall, generating a point cloud representative of the object, albeit sometimes pierced. Much better performing were the ground-based systems: both TLS and MMS allowed the laser scanner to be positioned/moved around the building, choosing the best vantage points or paths for the specific task. Indeed, a limitation of these terrestrial tools lies in the inability to acquire the higher portions of the buildings, thus generating incomplete object geometries. 

Regarding the technical performances of the three LiDAR models, the DJI Zenmuse L1 has the lowest performance in terms of level of detail, density, number of points, and related noise. Although the Kaarta Stencil MMS performs less well than Faro Focus TLS in terms of the density and resolution of the acquired data, it manages to compensate because of the acquisition methodology, which allows user movement to increase/decrease the density and amount of acquired data. On the other hand, the MMS lacks an imaging sensor able to colorize the point cloud, while L1 has visible images that also allow photogrammetric survey.

Concerning the processing, the MMS dataset was the fastest in processing time thanks to the high-level SLAM and loop closure algorithm implemented in the post-processing routine. Moreover, the ALS dataset can be processed relatively fast and, thanks to the UAS RTK positioning, automatically. The TLS required both more human effort and processing time (for point cloud registration, filtering, and segmentation). 

The resulting point clouds have been analyzed and compared, focusing attention on the number of points acquired by the different systems, the density, and the nearest neighbor distance. The TLS survey is more accurate and provides a higher number of points, but the overall mean density is less than the MMS cloud. The ALS is less dense and has almost 30 times fewer points than the TLS data. 

The *M3C2* cloud-to-cloud distance computation algorithm has been applied to compare MMS and ALS with the TLS used as a ground truth. The results highlight that the direct geo-referencing of the RTK positioning performed by the UAV receiver is not enough to obtain reliable data as the mean difference between the reference TLS cloud (geo-referenced with the geodetic GNSS survey) and the ALS cloud is about 30 cm. Therefore, a ground GNSS survey should always be performed using markers or reflective targets visible from the DJI L1 sensor to be used as a GCP. Despite this, geo-referencing performed on the fly can be an excellent starting point to quickly apply co-registration algorithms. In this work, ICP was applied to aerial survey results, resulting in an average distance between point clouds of less than 1 cm. The L1 sensor can exploit the three different return signals to acquire data in forestry and densely vegetated areas. In this work, it has been observed that the system is able to gather data beneath the canopy, also reaching that portion of the building.

## Figures and Tables

**Figure 1 sensors-22-06314-f001:**
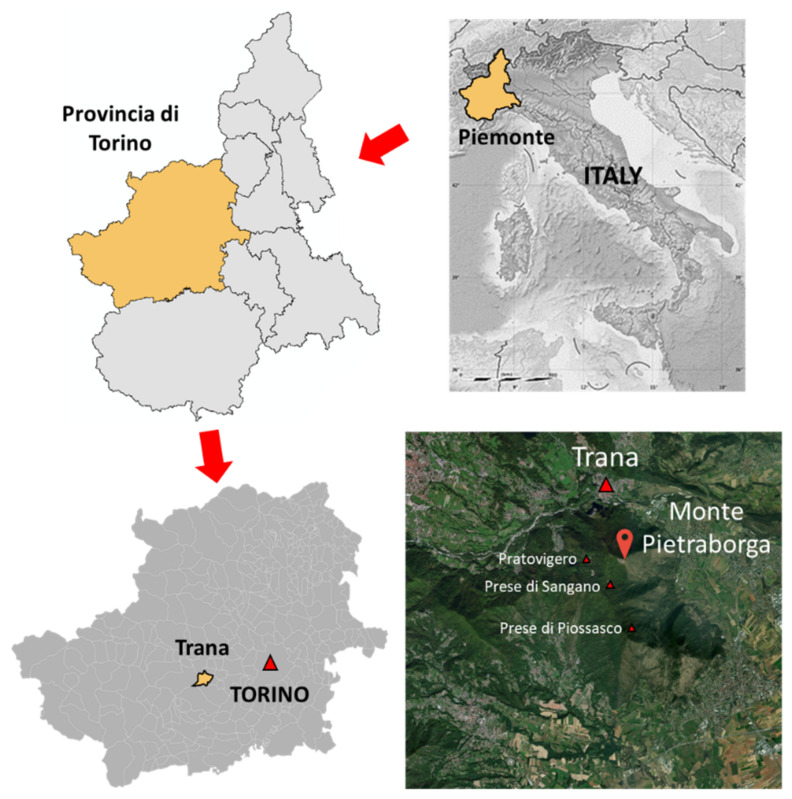
Monte Pietraborga location (Trana, Provincia di Torino, Piemonte, Italy).

**Figure 2 sensors-22-06314-f002:**
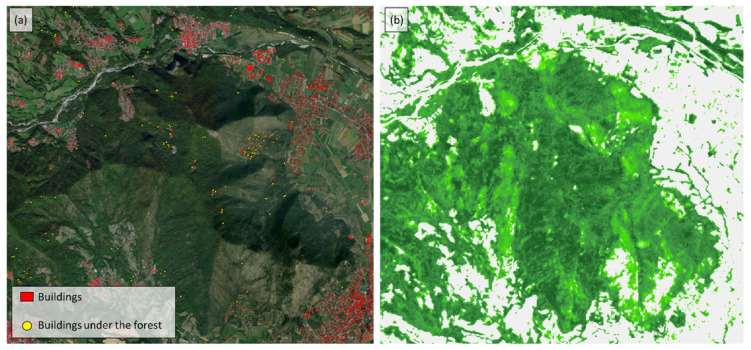
(**a**) In red, the surface of the buildings registered in the cadastre of the Città Metropolitana di Torino. In yellow, the centroids of the buildings that are covered by vegetation; (**b**) Tree cover density, dominant leaf type, and forest type products for reference year 2018 in 10 m resolution (High-Resolution Layers—Forest).

**Figure 3 sensors-22-06314-f003:**
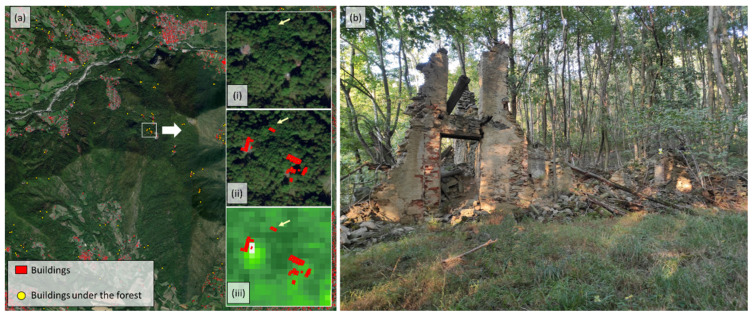
(**a**) Location of the heritage element chosen for testing the different geomatic sensors (37,5016.508 East; 4,986,232.369 North—WGS 1984, UTM Zone 32 N). On the right and from top to bottom: (i) The first white box shows the orthophotography of the chosen site. As can be see it is completely covered by vegetation. (ii) In the second frame it is possible to see the overlapping layer of buildings, with the yellow dots of the survey done (where it indicates that the building is under a wooded area). (iii) The third frame shows the High-Resolution Layers-Forest layer, where the layer of the buildings and the layer of the dots under the forest have been overlaid.; (**b**) Image of the heritage element.

**Figure 4 sensors-22-06314-f004:**
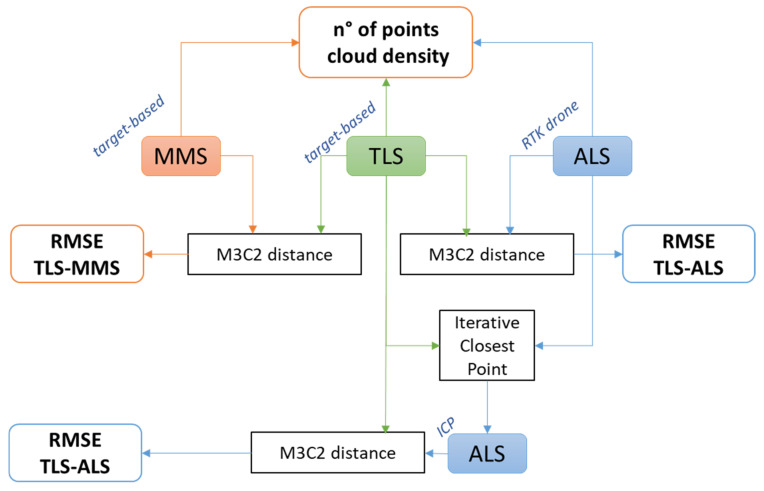
Validation analysis workflow to compare FARO Focus 3D X130 TLS, Kaarta Stencil 2–16 MMS, and DJI Zenmuse L1 ALS.

**Figure 5 sensors-22-06314-f005:**
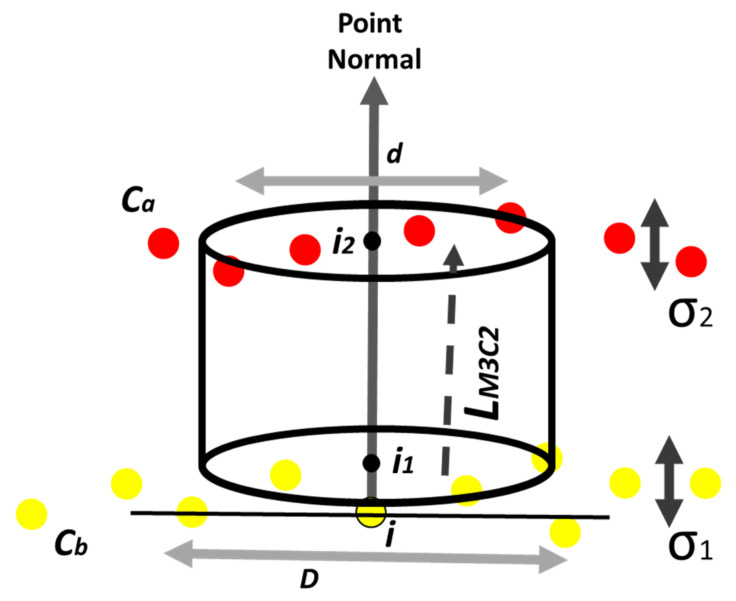
Cylinder projection distance *M3C2*. The point normal for *i* is calculated using the scale, *D*. A cylinder with a diameter d and a user-specified maximum length is used to select points in *Cb* and *Ca* (point clouds to be compared) for the calculation of *i_1_* and *i_2_*, respectively. *L_M3C2_* is the distance between *i_1_* and *i_2_* and is stored as an attribute of *i*. The local and apparent roughness of *Cb* and *Ca* are calculated as *σ_1_* and *σ_2_*, respectively, which are used to calculate the confidence interval of the spatial variable *i* [31,32].

**Figure 6 sensors-22-06314-f006:**
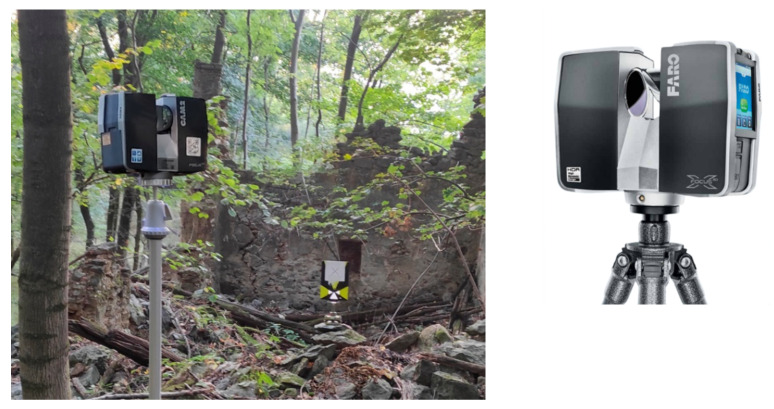
Faro Focus 3D X130 Terrestrial Laser Scanner device used for the detailed digitalization of the building and its environment. On the left, the laser scanner is shown during data collection and on the right the equipment in detail.

**Figure 7 sensors-22-06314-f007:**
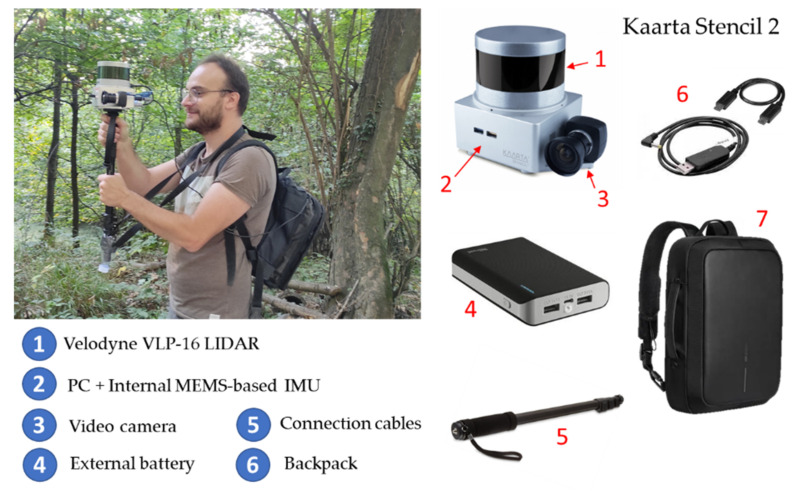
Kaarta Stencil 2–16 wearable mobile mapping system used: main components and a photo taken during the data acquisition.

**Figure 8 sensors-22-06314-f008:**
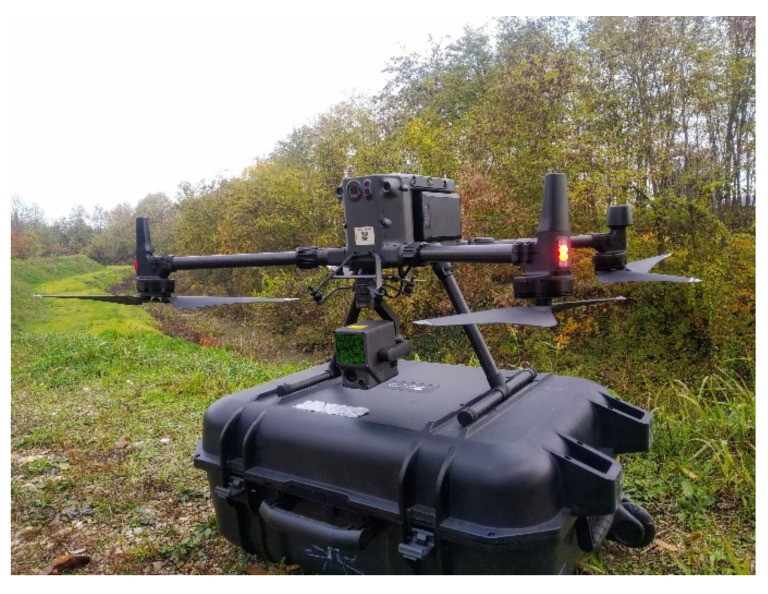
DJI Matrice 300 RTK with DJI Zenmuse L1 ALS.

**Figure 9 sensors-22-06314-f009:**
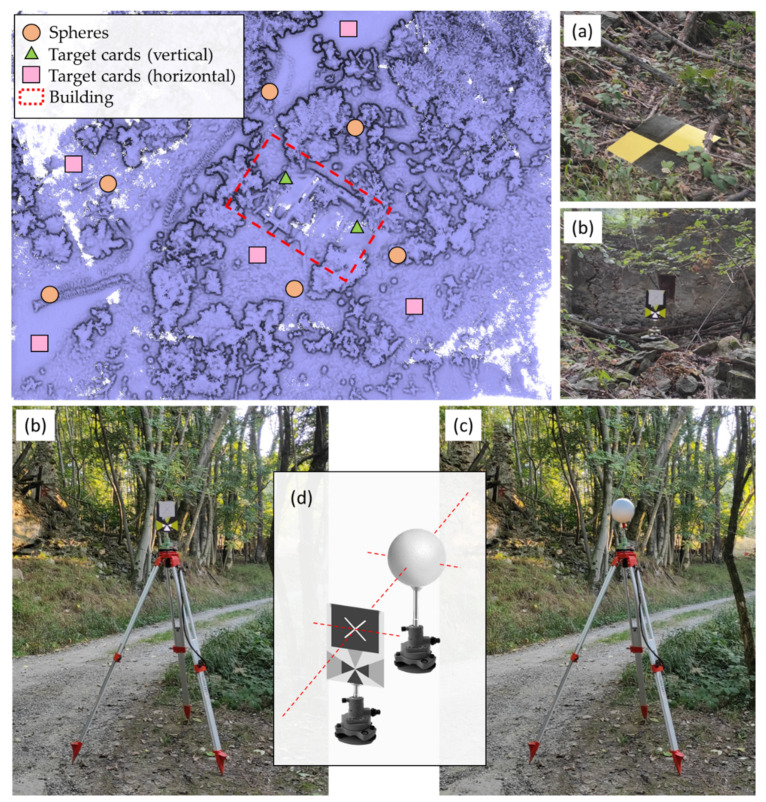
Network of registration spheres and target cards (horizontally and vertically), plant view. (**a**) Horizontal target cards; (**b**) vertical target cards; (**c**) spheres; (**d**) image of the calibration system.

**Figure 10 sensors-22-06314-f010:**
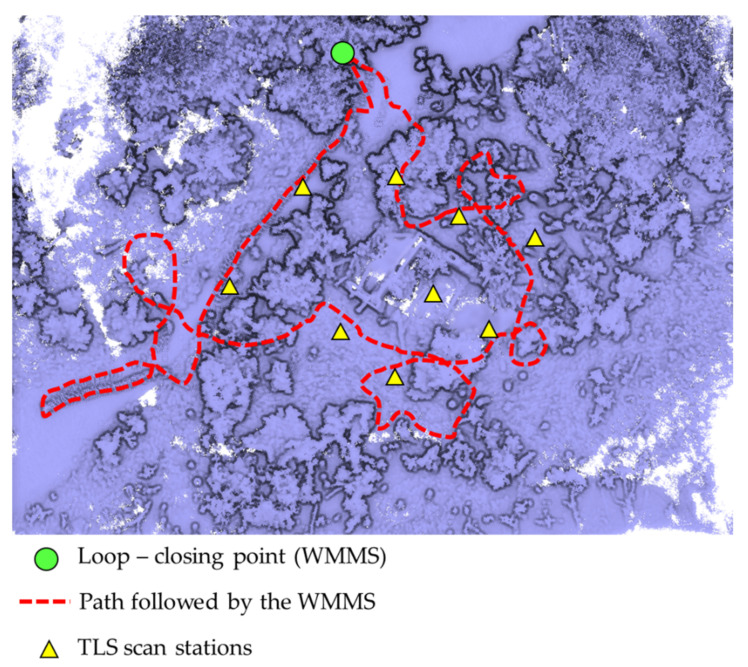
TLS scan stations and path followed during the data acquisition with the WMMS.

**Figure 11 sensors-22-06314-f011:**
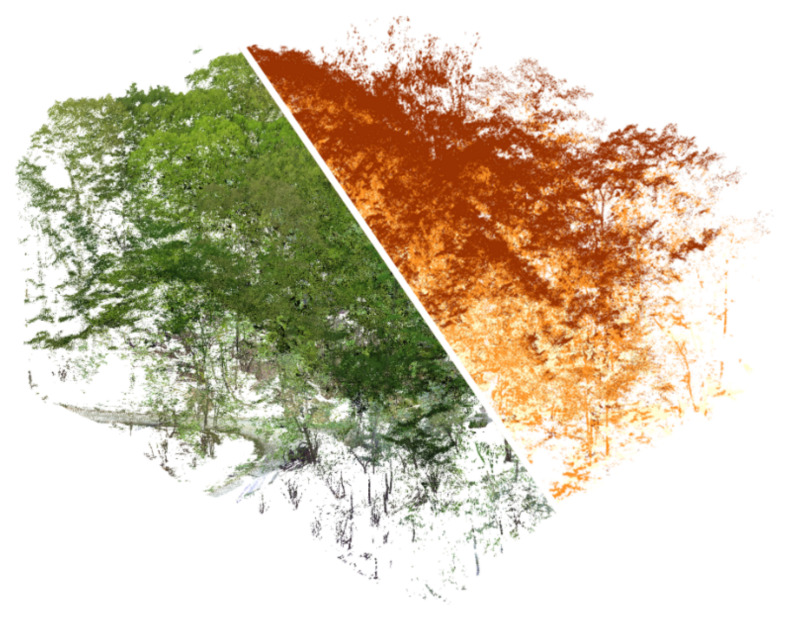
Point cloud acquired by the DJI L1 sensor, visualized in RGB (**left**) and in return number classification (**right**).

**Figure 12 sensors-22-06314-f012:**
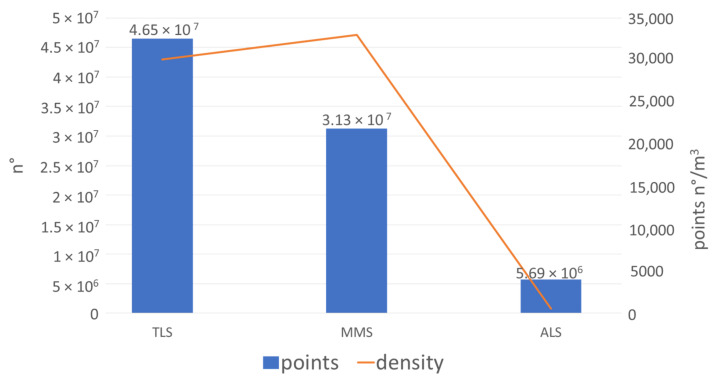
Number of cloud points for each original dataset acquired and volume density in 1 m^3^. All the data are extracted by an area of 3000 m^2^.

**Figure 13 sensors-22-06314-f013:**
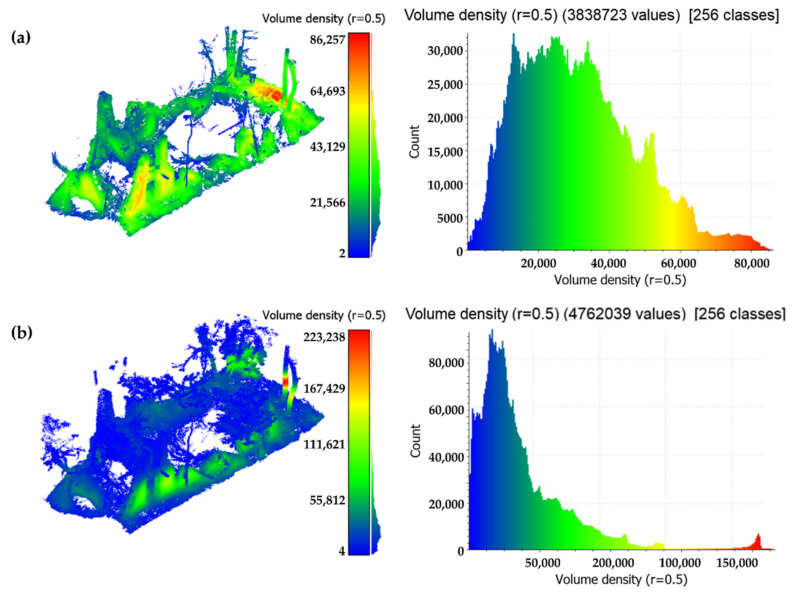
Point cloud density expressed in pts/m^3^ of TLS data and relative histogram (**a**) with respect to Kaarta Stencil WMMS data (**b**).

**Figure 14 sensors-22-06314-f014:**
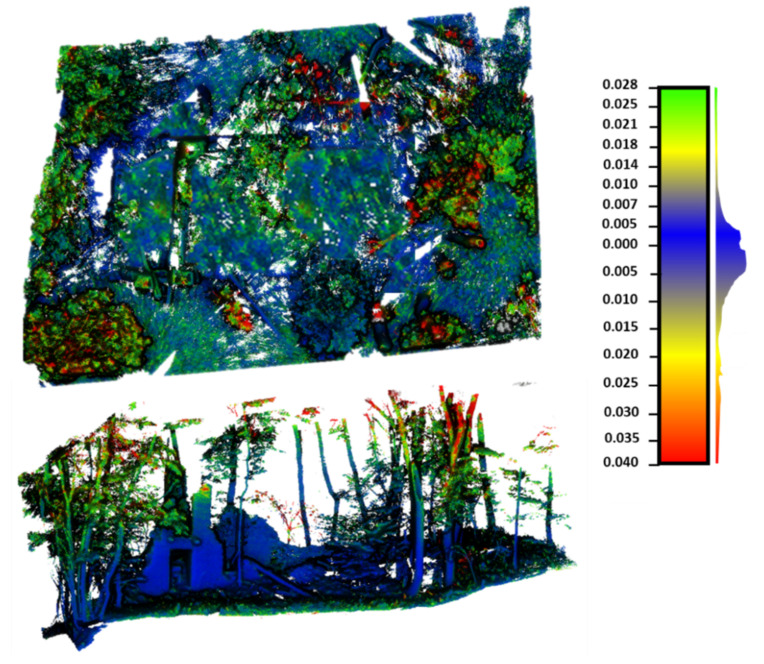
Discrepancies obtained from the comparison of the TLS and MMS point clouds. It is observed that most of the MMS point clouds show a blue color, indicating that the discrepancies between the MMSS and the TLS are under or near 1 cm. Yellow, red, and blue areas indicate vegetation zones (with minimal wind, they can move and generate discrepancies between point clouds).

**Figure 15 sensors-22-06314-f015:**
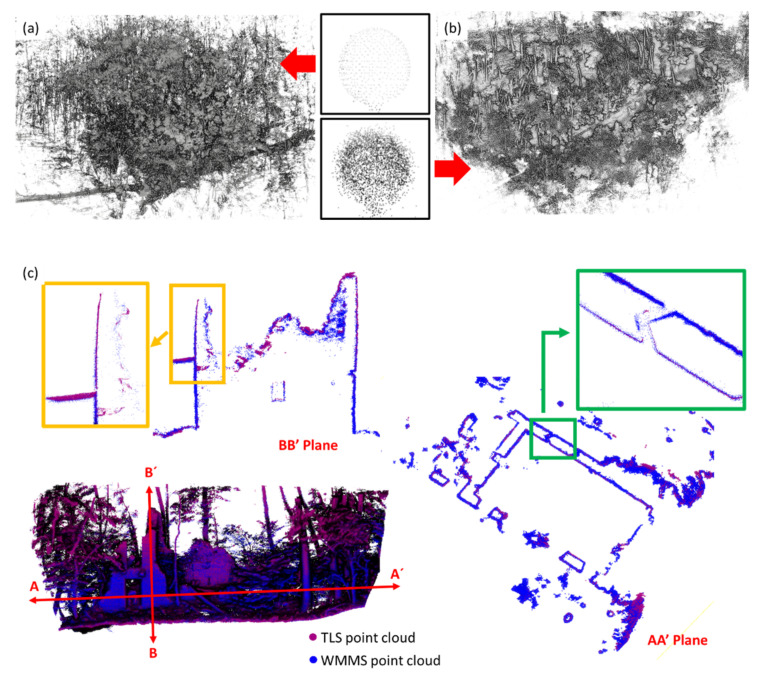
(**a**) TLS point cloud; (**b**) MMS point cloud; (**c**) comparison between the TLS and the MMSS point clouds—the TLS point cloud is in purple and the MMS point cloud is in blue. The AA’ Plane cuts the area of the building horizontally. BB’ Plane cuts the building vertically.

**Figure 16 sensors-22-06314-f016:**
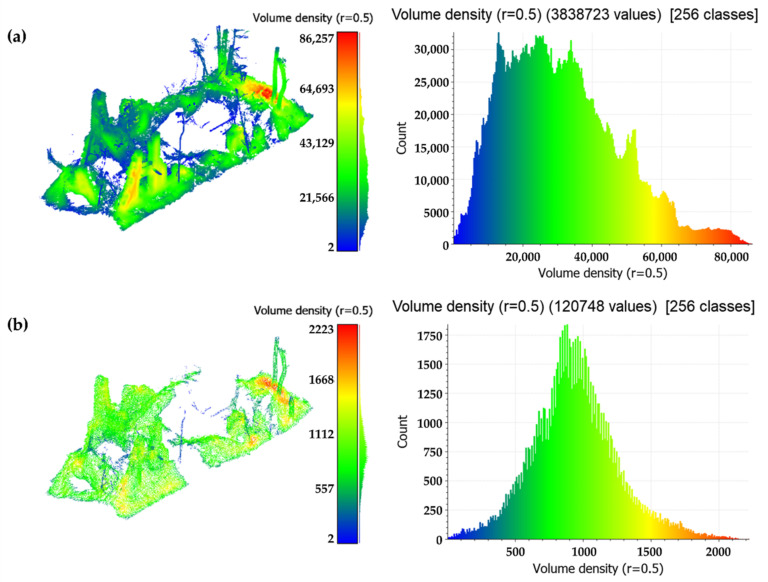
Point cloud density of TLS data and relative histogram (**a**) with respect to L1 data (**b**).

**Figure 17 sensors-22-06314-f017:**
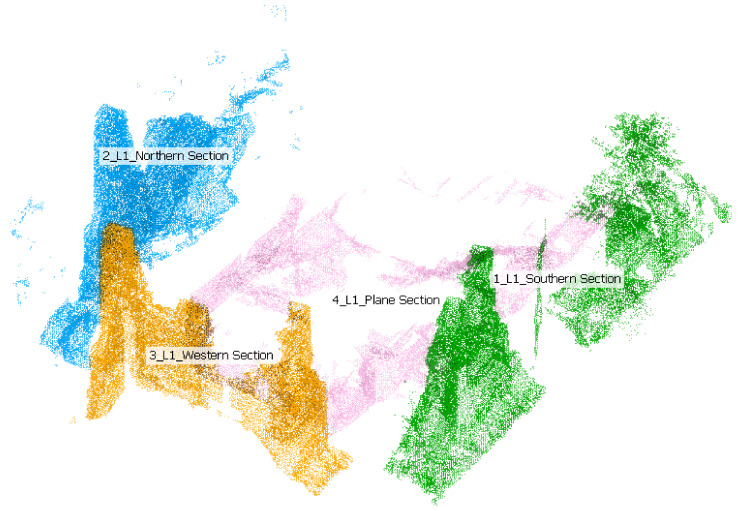
Region analysis for L1 point cloud comparison.

**Figure 18 sensors-22-06314-f018:**
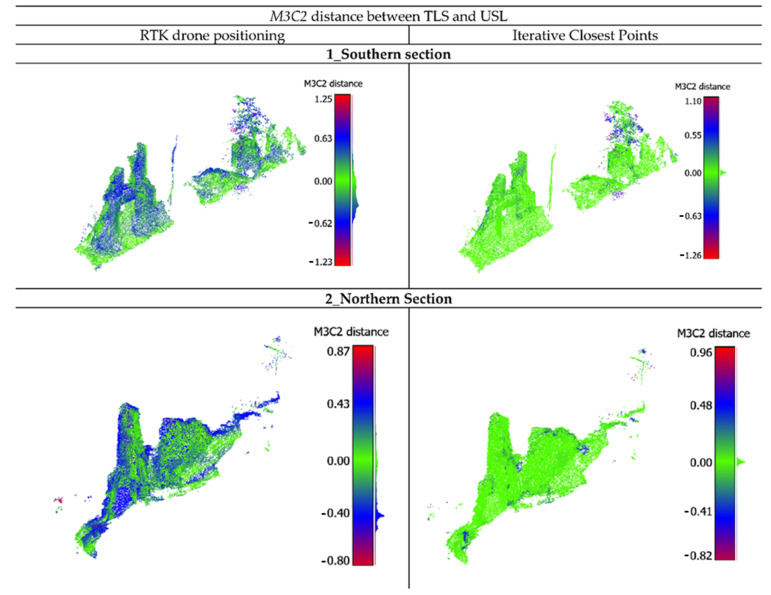
*M3C2* analysis of the building along different sections with direct geo-referencing and after ICP. Faro Focus 3D X130 TLS is used as reference and DJI Zenmuse L2 as compared dataset.

**Table 1 sensors-22-06314-t001:** Parameters used for running the *M3C2* algorithm. The first column shows the parameters suggested by the literature review, while the last column reports the tuned parameters used after some iterations.

*M3C2* Parameters		
*M3C2VER*	first	last
*NormalScale*	0.25	0.251872
*NormalMode*	0	1
*NormalMinScale*	0.125936	0.125936
*NormalStep*	0.125936	0.125936
*NormalMaxScale*	0.503744	0.503744
*NormalUseCorePoints*	true	false
*NormalPreferedOri*	0	6
*SearchScale*	0.25	0.2
*SearchDepth*	0.25	4
*SubsampleRadius*	0.05	0.125936
*SubsampleEnabled*	true	false
*RegistrationError*	0.003	0.003
*RegistrationErrorEnabled*	true	false
*UseSinglePass4Depth*	false	false
*PositiveSearchOnly*	false	false
*UseMedian*	false	false
*UseMinPoints4Stat*	false	false
*MinPoints4Stat*	5	5
*ProjDestIndex*	1	2
*UseOriginalCloud*	false	true
*ExportStdDevInfo*	true	true
*ExportDensityAtProjScale*	true	true
*MaxThreadCount*	16	16
*UsePrecisionMaps*	false	false
*PM1Scale*	1	1
*PM2Scale*	1	1

**Table 2 sensors-22-06314-t002:** Setting Kaarta Stencil 2 parameters input for outdoor environment.

**Resolution of the Point Cloud in Map File (m)**
*voxelSize* = 0.4 m
**Resolution of the Point Cloud for Scan Matching and Display (m)**
*cornerVoxelSize* = 0.2 m *surfVoxelSize* = 0.4 m *surroundVoxelSize* = 0.6 m
**Minimum Distance of the Points to Be Used for the Mapping (m)**
*blindRadius* = 2 m

**Table 3 sensors-22-06314-t003:** Basic characteristics of the UAV DJI Matrice 300 RTK.

Weight	Approx. 6.3 kg (with One Gimbal)
Max. transmitting distance (Europe)	8 km
Max. flight time	55 min
Dimensions	810 × 670 × 430 mm
Max. payload	2.7 kg
Max. speed	82 km/h
GNSS	GPS + GLONASS + BeiDou + Galileo
Accuracy in hovering flight (P mode, with GPS)	Vertical: ±0.1 m (vision system activated). ±0.5 m (GPS activated) ±0.1 m (RTK activated) Horizontal: ±0.3 m (vision system on) ±1.5 m (GPS on) ±0.1 m (RTK activated)
RTK positioning accuracy	With RTK activated and locked: 1 cm + 1 ppm (Horizontal) 1.5 cm + 1 ppm (Vertical)

**Table 4 sensors-22-06314-t004:** Basic characteristics of the DJI Zenmuse L1 laser scanner.

Dimensions	152 × 110 × 169 mm
Weight	930 ± 10 g
Maximum Measurement Distance	450 m at 80% reflectivity, 190 m at 10% reflectivity
Recording Speed	Single return: max. 240,000 points/s; Multiple return: max. 480,000 points/s
System Accuracy (1σ)	Horizontal: 10 cm per 50 m; Vertical: 5 cm per 50 m
Distance Measurement Accuracy (1σ)	3 cm per 100 m
Beam Divergence	0.28° (Vertical) × 0.03° (Horizontal)
Maximum Registered Reflections	3
RGB camera Sensor Size	1 in
RGB Camera Effective Pixels	20 Mpix (5472 × 3078)
IMU	Refresh date = 200 Hz Accelerometer range = ± 8 g

**Table 5 sensors-22-06314-t005:** Volume density and number of neighbors comparison in a volume of 1 m^3^ between the reference point cloud (Faro Focus 3D X130) and the compared one (Kaarta Stencil 2–16).

	**Volume Density (Radius = 0.5 m)**		
**LiDAR**	**N°. of Points**	**Mean (m)**	**St. Deviation (m)**
FARO TLS	3,838,723	30,943.20	16,340.60
Kaarta Stencil	4,762,039	42,419.30	38,283.40
	**Number of Neighbors (Radius = 0.5 m)**		
	**N°. of Points**	**Mean (m)**	**St. Deviation(m)**
FARO TLS	3,838,723	16,181.60	8577.41
Kaarta Stencil	4,762,039	22,210.70	20,045.10

**Table 6 sensors-22-06314-t006:** Volume density and number of neighbors comparison in a volume of 1 m^3^ between the reference point cloud (Faro Focus 3D X130) and the compared one (DJI Zenmuse L1).

	**Volume Density (Radius = 0.5 m)**		
**LiDAR**	**N°. of Points**	**Mean (m)**	**St. Deviation (m)**
FARO TLS	3,838,723	30,943.20	16,340.60
DJI L1	120,748	939.02	315.00
	**Number of Neighbors (Radius = 0.5 m)**		
	**N°. of Points**	**Mean (m)**	**St. Deviation(m)**
FARO TLS	3,838,723	16,181.60	8577.41
DJI L1	120,748	491.13	165.56

**Table 7 sensors-22-06314-t007:** Results of *M3C2* algorithm. Statistical values of the deviations between FARO Focus 3D X130 and DJI Zenmuse L1 (with direct geo-referencing and after ICP).

*M3C2* Distance					
Alignment	Nº. of Valid Points	Min (m)	Max (m)	Mean (m)	St. Deviation (m)
**Area of Interest**					
Direct Georef.	112,863	0.00	4.11	0.36	0.47
ICP	117,943	0.00	1.29	0.06	0.00
**1_Southern Section**					
Direct Georef.	34,034	0.00	1.25	0.27	0.18
ICP	34,960	0.00	1.25	0.07	0.14
**2_Northern Section**					
Direct Georef.	34,216	0.00	0.87	0.27	0.17
ICP	34,179	0.00	0.95	0.03	0.06
**3_Western Section**					
Direct Georef.	27,720	0.00	0.72	0.24	0.15
ICP	28,211	0.00	0.48	0.016	0.03
**4_Plane Section**					
Direct Georef.	28,985	0.00	1.21	0.27	0.17
ICP	29,035	0.00	0.98	0.05	0.11

**Table 8 sensors-22-06314-t008:** *M3C2* distance analysis between TLS and ALS for the building area. Mean distance for 5%, 50%, and 95% of the points.

*M3C2* Distance Analysis			
Alignment	5%	50%	95%
**Area of Interest**			
Direct Georef.	0.02	0.24	1.1
ICP	0.00	0.01	0.07
**1_Southern Section**			
Direct Georef.	0.02	0.26	0.57
ICP	0.00	0.02	0.37
**2_Northern Section**			
Direct Georef.	0.02	0.28	0.51
ICP	0.00	0.01	0.15
**3_Western Section**			
Direct Georef.	0.02	0.24	0.46
ICP	0.00	0.01	0.06
**4_Plane_Section**			
Direct Georef.	0.02	0.27	0.53
ICP	0.00	0.01	0.28

**Table 9 sensors-22-06314-t009:** Density and point number statistics of different L1 signal returns.

	N°. of Points	N°. of Neighbor (r = 0.5 m)		Density (r = 0.5 m)	
		Mean (m)	St. Dev (m)	Mean (m)	St. Dev (m)
Beam 1	3,476,674	229.47	157.33	438.25	300.49
Beam 2	1,976,398	150.57	118.80	287.58	226.89
Beam 3	654,926	125.98	105.84	240.61	202.14

## Data Availability

Not applicable.

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
