# Peer review of "Evaluation of Different LiDAR Technologies for the Documentation of Forgotten Cultural Heritage under Forest Environments"

_sensors, 2022, doi:10.3390/s22166314_

Round 1

Reviewer 1 Report

The topic dealt with is undoubtedly topical in the landscape of integrating data from multiple sensors. The state of the art is quite thorough and the exposition is clear. However, minor modifications are recommended. In some images, such as 12, 15, and 17, reading the data in the graphs is particularly difficult because they are too small. We also recommend reporting the units of measurement for clarity. Another observation concerns the colour gradients used to display the results of the comparison with M3C2. You might consider calibrating the gradient transition values better to make the results more readable and comparable effectively. Another tip is to elaborate, possibly in a separate section, on how the M3C2 algorithm itself was calibrated and how the various parameters governing it were set. Finally, minor corrections are recommended: Line 20 - “accuracies”; Line 85 - “search for”; Line 107 - “resolution”; Line 115 - “allow installing”; Line 184 - “in search for”; Line 187 - “in a very poor”.

Author Response

Authors thank this referee for the high quality of this new review and all the hints and remarks identified. Manuscript has been reviewed and improved following your advice and hints.

Reviewer 2 Report

The paper presents the comparison of three different LiDAR technologies (TLS, MMS and drone ALS) in the case of the documentation of cultural heritage under forest environments. The topic is current and of interest, and responds to a specific issue.

Among the three systems used, the interest in comparing the ALS system with the other two stands out, as it is a more recent technology (although the MMs is also quite new). Moreover, the choice of focusing the study on a specific environment (the forest) provides an added differential fact, dealing with the issues of working in these places.

The introduction gives a good context to the investigation. Regarding references, although the works are recent and related to the investigation, there are perhaps too many self-citations and also the references are repeated excessively throughout Section 1.

The methods and technical development of the work are adequate, and the results provide the necessary data to formulate the conclusions, which are coherent with the data presented.

Regarding figures, some of them would need minor edits:

- Figure 10 appears two times (pg. 14 and 15),

- Figure 12: the histogram of b) appears with smaller numbers than in a) and are hardly eligible.

- Figure 13: could perhaps be separated into two figures, or at least give the sections a better graphics.

- Figure 17: the size of numeric values is not consistent across sections

The writing and structure of the text are clear in general terms. The explanation of the TLS vs ALS analysis could be improved in terms of the sentence on lines 538 and 539 and the need to divide into regions for the two analyses.

It would also be interesting to include a comment about the scanning time required by the three procedures, since only the time reference of the MMS appears in the text (12 min., line 412).

On the other hand, the use of comma or point to separate decimals have to be revised, since it is not consistent throughout the text (see the number of points in lines 399, 419 and 439, for example).

Finally, some typos and faults have been identified in the text. For example: line 476 “an high-tiled”; line 451 “an higher”; line 582 “analuzyed”. 

Author Response

Authors thank this referee for the high quality of this revision and all the hints and remarks identified. Manuscript has been reviewed and improved following your advice and hints. All your considerations have been added to the new version of the manuscript.
